# Correctional Officers’ Health Literacy and Practices for Pulmonary Tuberculosis Prevention in Prison

**DOI:** 10.3390/ijerph191811297

**Published:** 2022-09-08

**Authors:** Suwida Rakpaitoon, Sasithorn Thanapop, Chamnong Thanapop

**Affiliations:** 1Master of Public Health Program, School of Public Health, Walailak University, 222 Tambon Thai-Buri, Thasala District, Nakhon Si Thammarat 80160, Thailand; 2Workers Health Research Center, Walailak University, 222 Tambon Thai-Buri, Thasala District, Nakhon Si Thammarat 80160, Thailand

**Keywords:** correctional officers, health literacy, pulmonary tuberculosis, prevention, prison

## Abstract

Tuberculosis (TB) prevention in prisons remains a problem that requires advocacy and partnership action. A correctional officer (CO) is responsible for enforcing the rules and maintaining routines at a prison and has the authority to support TB prevention under the limitations of health manpower in prisons. The objective of this cross-sectional study was to determine the health literacy (HL) and practices of TB prevention and their association among Thailand’s COs. A total of 208 COs participated using a random sampling method. A self-administered questionnaire on HL and TB prevention practices was used for data collection. Descriptive statistics, Pearson’s chi-square test, and binary logistic regression were used for the association analysis. The majority of the participants were male (71.2%), married (60.1%), had a bachelor’s degree (60.6%), and had never been trained in TB prevention (90.9%). In total, 63.0% had adequate HL, whereas 78.4% had good practices, and this corresponded with personal prevention (75.5%) and work prevention (74.6%). Significant associations were identified for education, and communication, decision-making, and self-management skills (*p* < 0.05). The probability (adjusted odds ratio [95% CI]) of good practices was higher among participants with adequate communication skills (7.92 [2.15–29.24]), adequate decision-making skills (6.00 [1.86–19.36]), bachelors’ degree or higher-level education (3.25 [1.12–9.39]), and adequate self-management skills (2.95 [1.08–8.11]). The study findings show that most of the COs have adequate HL which is associated with good practices in TB prevention. Prisons should support HL development among COs for partnership and sustainable TB prevention under the constraint of health personnel.

## 1. Introduction

Tuberculosis (TB) is a contagious disease that constitutes a significant public health problem and is still one of the major causes of preventable death in the world. TB is an infectious disease caused by TB bacteria (Mycobacterium tuberculosis) and mostly affects the lungs. It can cause serious damage to the lungs and other organs [1]. When a person develops active TB disease, the symptoms may be mild for many months. This can lead to delays in seeking care, and results in transmission of the bacteria to others [1]. In 2020, the 30 high TB burden countries accounted for 86% of new TB cases. Eight countries account for two-thirds of the total, with India leading the count, followed by China, Indonesia, the Philippines, Pakistan, Nigeria, Bangladesh, and South Africa [1]. Globally, close to one in two TB-affected households face costs higher than 20% of their household income, according to the latest national TB patient cost survey data [1]. The World Health Organization (WHO) had estimated that, by 2020, the global incidence (new and recurrent) of TB would reach 9.9 million (127 per 100,000 persons) [2]. In 2021, the WHO listed three new high-TB countries for 2021–2025. Though, no longer among the top 30 countries with the highest burden of multidrug/rifampicin-resistant TB (MDR/RR-TB; Group 3), Thailand is listed among countries with a high TB burden (Group 1) and TB/HIV burden (Group 2) [2]. Thailand has the seventh highest and highest number of prisoners worldwide and among the Asian countries, respectively [2]. TB is a major infectious disease in the prison system, and evidence has demonstrated that, on average, the risk of developing TB in prisoners is 6–30 times greater than in the general population [3,4,5]. In Thailand in 2019, 4338 of the 354,905 prisoners registered for TB treatment, 409 were infected with HIV or had AIDS, and 77 had MDR/RR-TB; this indicated that TB spread in prisons remains a problem that requires intensive disease-control measures [6]. Furthermore, factors that increase TB-related morbidity and mortality include a higher prison population, delayed legal processes, inadequate nutrition and access to health services, overcrowded spaces, poor ventilation, violence, and TB concurrent with HIV/AIDs [7,8]. Moreover, the burden of TB in incarcerated populations raises substantial concerns about the spread and control of other infectious diseases—particularly the coronavirus disease (COVID-19)—in such settings [9,10,11].

The first strategy of the Thailand National Action Plan against Tuberculosis 2017–2021 (Additional 2022) is to expedite the investigation of TB cases by screening high-risk groups, including prisoners [2,12]. Rapid access to diagnostic services requires equipment and personnel to support operations as well as the creation of operational guidelines for the health service unit [13,14]. However, the restrictions that exist in prisons in terms of providing health services with one prison nurse per 1250 prisoners probably affected the effectiveness of healthcare delivery and TB prevention [15,16,17]. Moreover, prisoners are a high-risk population because of the increased risk of exposure and reduced access to quality TB services [5]. The correctional officers (COs) are uniformed law enforcement officials responsible for the custody, supervision, safety, and regulation of prisoners. They are also responsible for enforcing the rules and maintaining routines at a prison, have the authority to support the health, environment, and welfare of the prisoners [17,18,19], and could play an important role in public health practice collaboration, such as TB prevention advocacy for prisoners, with prison nurses [6]. Moreover, COs are empowered and knowledgeable agents who can ensure health delivery through a partnership with public health personnel in prisons. As a result, correctional officers could use their knowledge, skills, and experiences to communicate health information about prisoners for understanding the gaps and to facilitate decision-making on appropriate health behavior.

Sustainable Development Goals (SDGs) Target 3.3 includes ending the TB epidemic by 2030 [20]. The End TB Strategy, adopted by the World Health Assembly in 2014, was intended to end TB as a global public health threat by 2035 [21]. The WHO is working closely with countries, partners, and civil society in scaling up the TB response using six core functions, including shaping the TB research and innovation agenda, stimulating the generation, translation, and dissemination of knowledge, setting norms and standards on TB prevention and care, promoting and facilitating their implementation, and developing and promoting ethical and evidence-based policy options for TB prevention and care [20]. Thus, under the limitations of health personnel for TB prevention in prisons, COs can participate to promote TB prevention and control activities in prisons together with prison nurses. This public health role of the CO will give at-risk prisoners rapid access to TB assessment and treatment. Thus, it could be an advocacy and partnership strategy for preventing and controlling TB in prisoners. Nevertheless, to advocate for and participate in TB prevention in prison, COs should have adequate health literacy (HL) to manage healthy behavior in TB prevention and control practices [1,22]. HL represents the cognitive and social skills that determine the motivation and ability of individuals to gain access to, understand, and use information in ways that promote and maintain good health [23,24,25]. In addition, Edwards et al. (2012) reported that HL can be acquired or contribute to one’s capacity to become health literate to manage health conditions by accessing information and availing of health services [6]. It is possible to negotiate and ensure access to appropriate treatment in order to provide group members with better knowledge and inculcate self-management skills [25]. Therefore, CO’s health literacy probably assists the prisoners in effectively achieving healthcare access despite the limitations of health personnel. Improving the HL skills among COs will also promote potential TB prevention and control of the prisoners. Previous studies in prison settings have focused on the assessment of the prevalence and systematic screening of TB by health personnel, but few studies have investigated the HL of correctional officers on TB prevention and control activities [4]. Therefore, this study was undertaken among Thai correctional officers to determine the HL and practices of TB prevention and control, as well as their associations. The study’s findings will enable understanding CO’s health literacy skills and practices on TB prevention and control and improving the skills to assist TB healthcare access through the advocacy and partnerships strategy in prisons under the constraints of health personnel.

## 2. Materials and Methods

### 2.1. Study Design and Participants

From December 2021 to February 2022 we conducted a cross-sectional study in five prisons in Nakhon Si Thammarat province, southern Thailand. The study population included 436 COs, and the sample size was calculated by using the finite population [26]; the proportion formula with *p* = 0.5 was 193; adjustment to ensure an additional 10% study population yielded a minimum sample size of 213. Stratified random sampling proportional to the size of each prison was used to select the participants [27].

### 2.2. Data Collection and Measurements

We used a questionnaire developed by the researchers for collecting information, including demographics, history of training on epidemic prevention and control, and HL and practices of TB prevention and control in prison. Data were collected through a self-administered questionnaire and data collection procedures were conducted by the researchers. Three measurements were employed to assess the overall quality of the questionnaire. The index of item objective congruence yielded a value of 0.7–1.0 for content validity of all parts, the KR 20 was 0.86 for cognitive skill, and the Cronbach’ alpha coefficient was 0.94 for the reliability assessment. Before data collection, the researcher provided the participants with an information sheet and a separate factsheet for better understanding of the background of this study, the objectives and the benefits, and to solicit their participation in the study.

### 2.3. Health Literacy Assessment

HL assessment about TB prevention and control in prisons was undertaken with a tool that was constructed by the authors using the concept of six aspects of HL skills. [24,28,29]. The HL assessment comprised six aspects, including (1) cognitive skills; (2) access skills; (3) communication skills; (4) decision skills; (5) self-management skills; and (6) media literacy, which were captured using 41 questions. The questions on cognitive skills about the knowledge of TB infection were structured (17 items), pre-coded, and mainly evoked dichotomous responses (‘*yes*’ or ‘*no*’). In addition, participants self-reported ‘*the truth about yourself for TB prevention and control*’ for the remaining five skills (24 items). The five-point response scale comprised scores which were applied to the five skills assessment as follows: (5) Extremely; (4) Quite a bit; (3) Somewhat; (2) A little bit; and (1) Not at all. Examples of these questions include, ‘*I sought information about TB prevention and control in prisons from the prison infirmary*’.

### 2.4. Practices for TB Prevention and Control in Prison

We asked the participants about what they were doing or would do to protect themselves from (16 items) and control TB transmission (12 items). The questions pertained to general personal TB prevention practices and hygiene, TB screening access, sanitation practices in prison, and caring for inmates with TB following the disease control measures in prisons [15]. For example, ‘*You avoid close contact with people who have flu-like symptoms or severe acute respiratory infections*’. The response options were: (5) Extremely; (4) Quite a bit; (3) Somewhat; (2) A little bit; and (1) Not at all.

### 2.5. Data Processing and Analysis

The data collected were entered into Epi data version 3.1 (EpiData is released as freeware by the non-profit organisation “The EpiData Association” Odense, Denmark, 2004) for cross-checking and exported to SPSS version 23.0 (IBM Singapore Pte Ltd., Singapore, Registration No. 1975-01566-C, IBM Order Reference Number: 63430128, IBM Customer Number: 537100) for analysis. Responses on cognitive skill (knowledge of TB infection) were positively aligned to ensure that the correct answer was scored 1 and the wrong answer was scored 0, and the scores were summed to create a total score in the range of 0 to 17. Similarly, the other skills of HL were calculated by summing the total score (range, 24–120). The score for practices ranged from 28 to 140. For each construct, we categorized a participant with a score below 80% of the possible maximum score as possessing ‘inadequate skill’; a score above the 80% score indicated ‘adequate skill’ which was based on a modified Bloom’s cut-off point and has been recently used in other studies [30]. Scores for practices ranged between 24 and 140. Moreover, we categorized practices into three levels according to the cut-off points, wherein a score of 80% and above, 60–79.9%, and less than 60% indicated good, moderate, and poor levels, respectively.

We calculated the means with standard deviations for quantitative variables and frequencies for categorical variables to obtain proportions. We applied the chi-square test (χ^2^) to assess the association between a pair of categorical variables. In the multivariable analyses we used binary logistic regression analyses to quantify the associations among demographics, HL, and practices [31]. Independent variables were included in the binary logistic regression model according to their significance in the bivariate analysis (*p* ≤ 0.2) and their lack of collinearity. We calculated the adjusted odds ratio (adj. OR) and 95% confidence intervals (CI) as a measure of the strength of the association. In the bivariate and multivariable analyses the level of significance was set at 5%.

### 2.6. Ethics Approval

This study was approved by the Human Research Ethics Committee of Walailak University, Thailand (no. WUEC-21-293-01). All methods were performed following the relevant guidelines and regulations set out in the Declaration of Helsinki. Written informed consent was obtained from all participants before their enrolment in the study. Participants were informed that participation was voluntary, confidential, and the results would remain anonymous.

## 3. Results

### 3.1. Description of the Participants

The study recruited 208 participants (97.65% recruitment rate) for the analysis. Missing data collection was due to incomplete questionnaires. In Table 1, we present the distribution of the participants by background characteristics. The majority of this cohort were men (*n* = 148; 71.2%), were married (*n* = 125; 60.1%), and their highest educational level was a master’s degree; however, most of the participants (*n* = 126; 60.6%) had a bachelor’s degree. Almost all participants (*n* = 169; 81.3%) had not been diagnosed with non-communicable diseases. The majority, 189 (90.9%), of participants responded ‘*never*’ to the question on the history of training in TB prevention and control and 72 (34.6%) had 1–4 years of work experience.

### 3.2. Health Literacy for TB Prevention and Control and Participants’ Knowledge on TB

Most participants had an adequate level of total HL score (*n* = 131; 63.0%). The average score for cognitive skills of TB infection-related knowledge was 12.87 of 17 (SD = 1.47). In total, 64 (30.8%) had an adequate level of knowledge. For each dimension of HL, at least half of the participants had adequate levels of HL, self-management skills (*n* = 150; 72.1%), media literacy skills (*n* = 138; 66.3%), decision-making skills (*n* = 132; 63.5%), and access skills (*n* = 113; 54.3%). However, those with adequate HL on communication skills was only 47.1% (*n* = 98) (Table 2).

### 3.3. Practices for TB Prevention and Control

Among our participants, 163 (78.4%) had good practices in overall TB prevention and control, and this corresponded with personal prevention and prevention of TB transmission in prison. TB prevention practices were at the good level for personal prevention (*n* = 157; 75.5%) and preventing the TB transmission (*n* = 155; 74.6%). The highest proportion of practices that were reported as being ‘*extremely*’ included ‘*wearing a mask while working*’ (*n* = 146; 70.2%) and ‘*wearing a mask while caring for TB patients*’. Nevertheless, only 34.1% had ‘extreme practices’ in ‘screening for tuberculosis every six months’ (data not shown in Table) (Table 3).

### 3.4. Associations between TB Prevention and Control Practices and Demographics with HL Dimensions

The analysis highlighted that all HL skills, except cognitive skills, were significantly associated with TB prevention and control practices (*p* < 0.05). We found a higher proportion of good practices in each HL dimension (Table 4). Next, the simple logistic regression found significant associations with education, communication, decision, and self-management skills (*p* < 0.05). The probability (adj. OR [95% CI]) of good practices was higher among participants with adequate communication skills (7.92 [2.15–29.24]), adequate decision-making skills (6.00 [1.86–19.36]), bachelor’s degree or higher-level education (3.25 [1.12–9.39]), and adequate self-management skills (2.95 [1.08–8.11]) (Table 5).

## 4. Discussion

TB is a persistent disease among prisoners; however, resource scarcity and a shortage of health personnel limit the applicability of TB prevention and control strategies in the prisons of developing countries [32,33,34]. The use of peer education to train prison inmates on health-related issues could be a very cost-effective alternative and has already been shown to improve screening and prevention in prisons [35,36,37]. In Thailand, the Department of Health Service Support (2018) trained volunteer prison inmates to gain competence in health, HL, and health behavior regarding surveillance, control, infectious disease prevention, and health problems [38]. In practice, discontinuity was found in operations, especially for TB healthcare. The COs probably perform TB healthcare during their duty in collaboration with the prison nurse. The study results of CO’s health literacy and prevention practices will be used to set strategic planning in TB prevention, utilizing the concept of health advocacy and participation of the COs and health personnel in prisons.

The majority of the participants had adequate overall health literacy for TB prevention and control. HL is a combination of reading and listening skills, data analysis, decision-making, and the ability to implement these skills during necessary health situations [39]. However, approximately seventy percent of the participants had inadequate cognitive skills. From the above-mentioned results, we inferred that most of the CO had a bachelor’s degree or higher educational level, could contribute more to society, and could understand and apply advanced steps for control of practices [32]. Nevertheless, the high percentage of inadequate TB prevention knowledge is probably related to the lack of training in epidemic prevention and control, of which only ten percent of the participants received training. This deficit in training might have affected the understanding of renewed knowledge and valid information about TB prevention. Therefore, discontinuing training programs would disrupt an individual’s cognitive skills. On the contrary, it has been noted that even individuals with adequate general literacy might not have adequate healthcare knowledge because the literacy demands in the context of healthcare are frequently more complex [40]. Cognitive skill is partly HL, and it could be developed through health educational interventions [39,41], thus, continuing training will increase the cognitive skills based on improving knowledge.

Concerning access skills, approximately half of the participants had adequate skills; however, there was inadequate access to information on TB prevention in prisons. The current shortage of prison nurses imposes a limitation on health manpower, including healthcare activities and information. Thus, it probably resulted in COs with inadequate skills for accessing TB healthcare information. The limited accessibility of TB prevention services and information reflected the vulnerability and need for service improvements in prisons [5,33].

The communication skills were adequate for more than half of the participants. Health communication consists of interpersonal or mass communication activities focused on improving the health of individuals and populations [42]. In practice, the operations undertaken by COs emphasized the control of prisoners, and these officers do not have specific qualifications in health risk communication and education. Therefore, TB prevention communication skills were limited according to the COs’ principal role.

However, COs are personal resources that encourage the prisoners to exercise control over their health as well as health-related social and environmental factors. Thus, developing health education and communication skills among the COs could facilitate various health competencies at both individual and community levels [25].

Notably, most of the participants had adequate media literacy, decision-making skills, and self-management skills, which comprised the highest proportion of all HL skills. The participants intended to implement self-care and transmission prevention, which corresponded to a proportion of good practices of more than seventy per cent. These results confirmed that HL is a means for enabling individuals to exert greater control over their health as well as over the range of personal, social, and environmental determinants of health [42]. However, this study was conducted during the COVID-19 pandemic, and most of the TB prevention activities correlated with those for COVID-19 prevention. Thus, more information on the infectious disease crisis and public health measures probably affected the concentration of decisions and practices among the participants. Moreover, the adequate HL level of most participants indicated that HL involves the development of relevant personal knowledge and capability as well as interpersonal and social skills. People with better HL will have skills and capabilities that enable their engagement in a range of health-enhancing actions and personal behaviors and the capability to influence others toward healthy decisions [43]. The COs can then efficiently provide and expand their health literacy to the prison inmates.

The final analysis established that the academic status and communication, decision-making, and self-management skills of HL were associated with TB prevention practices of the participants. This finding corresponds with the perspective of Abel and McQueen, who clarified that critical HL skills, composed of decision-making and self-management skills, are essential for establishing high expectations for preserving health, especially against infectious diseases such as COVID-19 [44]. Moreover, the good practices of TB prevention among the COs were probably due to the perception and motivation of public health measures amidst the COVID-19 pandemic to make and appropriately manage self-health decisions. In Thailand, the COVID-19 prevention measures in prisons are consistent with the tuberculosis prophylaxis strategy [45]. Thus, the predominant relationship between academic status, adequate critical skills of the HL, and practices for TB prevention reflect the potential readiness of health advocators among the COs. Although there was no statistically significant association between access skills, media literacy, and prevention practices, this was probably caused by the dominant role of the CO in rule and prison enforcement, and were not directly relevant to TB practices.

Practical implications of this study are that COs require health education programmes to improve health literacy skills for TB health services and prevention, especially cognitive, communication, and access skills. The adequate HL of correctional officers also focuses not only on the cognitive principles of comprehending, analyzing, and applying health information to decisions about health, but also on the social skills involved in those interactions with other people and organisations that are necessary for transforming decisions into practice [42]. Thus, health literacy improvement of COs is a salient factor for effectively providing TB prevention and control among prisoners under the constraint of health personnel in prisons.

### Limitations

There are some limitations of this study. Due to the prevailing COVID-19 pandemic, we studied a representative sample of the prisons in the southern region of Thailand; therefore, our results may not be generalizable to the other regions in Thailand.

## 5. Conclusions

In conclusion, this study confirmed that most of the COs were well educated and had adequate health literacy for TB prevention. This requisite level of HL could lead to good practices for TB prevention in prisons, both at the individual and community level, as well as in prisons. The majority of COs had highly adequate interactive and critical HL skills and communication, decision-making, and self-management skills, and these were strongly associated with the practical application of TB prevention. Further studies should be conducted at the national level, which probably would comprise different study populations and should focus on peer education-based HL skill improvement programmes to strengthen the advocacy role and partnership of the COs for sustainable TB prevention in prisons.

## Figures and Tables

**Table 1 ijerph-19-11297-t001:** Characteristics of participants (*n* = 208).

Characteristics	Number (%)
Sex	
Male	148 (71.2)
Female	60 (28.8)
**Age**	
<35	59 (28.4)
35–44	83 (39.9)
45–59	66 (31.7)
Mean 40.4, SD 8.5, min 24, max 59	
**Marital status**	
Single/widow/separate	83 (39.9)
married	125 (60.1)
**Education status**	
Secondary school/diploma	41 (19.7)
Bachelor’s degree	126 (60.6)
Master’s degree	41 (19.7)
**Non-Communicable Disease**	
Absence	169 (81.3)
Presence	39 (18.7)
**History of training on epidemic prevention and control**	
Ever	19 (9.1)
Never	189 (90.9)
**Current working duration (year)**	
1–4	72 (34.6)
5–9	28 (13.5)
10–14	37 (17.8)
15–19	39 (18.8)
>20	32 (14.4)

**Table 2 ijerph-19-11297-t002:** Health literacy dimensions about TB (*n* = 208).

Dimensions	Skill Level: Number (%)	
	Inadequate	Adequate
Cognitive skill	144 (69.2)	64 (30.8)
Total of score 17; mean, 12.87; SD, 1.47; min, 8; max, 16		
Access skill	95 (45.7)	113 (54.3)
Media literacy skill	70 (33.7)	138 (66.3)
Communication skill	110 (52.9)	98 (47.1)
Decision skill	76 (36.5)	132 (63.5)
Self-management skill	58 (27.9)	150 (72.1)
Total of HL	77 (37.0)	131 (63.0)
Total of score 120; mean, 88.15; SD, 17.68; min, 42; max, 120		

**Table 3 ijerph-19-11297-t003:** Practices for TB prevention and control.

Practice	Practices: Number (%)		
	Poor (<60.0)	Moderate (60–79.9)	Good (≥80.0)
Personal prevention	1 (0.5)	50 (24.0)	157 (75.5)
Preventing the TB transmission	6 (2.9)	47 (22.6)	155 (74.6)
Overall	1 (0.5)	44 (21.2)	163 (78.4)

**Table 4 ijerph-19-11297-t004:** The bivariate association between TB prevention and control practices and demographic and HL dimensions.

Factors	Practices: Number (%)		*p*
	Poor-Moderate (*n* = 45)	Good (*n* = 163)	
**Sex**			0.462
Male	34 (23.0)	114 (77.0)	
Female	11 (18.3)	49 (81.7)	
**Age (year)**			0.244
<40	24 (25.3)	71 (74.7)	
40–59	21 (18.6)	92 (81.4)	
**Educational status**			0.185
Secondary school or diploma	12 (29.3)	29 (70.3)	
Bachelor’s and Master’s degree	33 (19.8)	134 (80.2)	
**Current working duration (y)**			0.902
<10	22 (22.0)	78 (78.0)	
≥10	33 (21.3)	85 (78.7)	
**Marital status**			0.501
Single/widow	16 (19.3)	67 (80.7)	
Married	29 (23.2)	96 (76.8)	
**NCDs**			0.293
Absence	39 (23.1)	130 (76.9)	
Presence	6 (15.4)	33 (84.6)	
**History of training on epidemic prevention and control**			0.217
Never	43 (22.8)	146 (77.2)	
Ever	2 (10.5)	17 (89.5)	
**Cognitive skill**			0.250
Inadequate	28 (19.4)	116 (80.6)	
Adequate	17 (26.6)	47 (73.4)	
**Access skill**			0.004
Inadequate	29 (30.5)	66 (69.5)	
Adequate	16 (14.2)	97 (85.8)	
**Media literacy skill**			<0.001
Inadequate	25 (35.7)	45 (64.3)	
Adequate	20 (14.5)	118 (85.5)	
**Communication skill**			<0.001
Inadequate	41 (37.3)	69 (62.7)	
Adequate	4 (4.1)	94 (95.9)	
**Decision skill**			<0.001
Inadequate	34 (44.7)	42 (55.3)	
Adequate	11 (8.3)	121 (91.7)	
**Self-management skill**			<0.001
Inadequate	26 (44.8)	32 (55.2)	
Adequate	19 (12.7)	131 (87.3)	

**Table 5 ijerph-19-11297-t005:** The binary logistic regression model of associations between demographic and HL dimensions with TB prevention practices.

Factors	Adj. Odds Ratio	95% CI	*p*
**Education** (bachelor-higher vs. secondary-diploma #)	3.25	1.12, 9.39	0.030 *
**Access skill** (adequate vs. inadequate #)	0.31	0.09, 1.01	0.051
**Media literacy skill** (adequate vs. inadequate #)	0.82	0.29, 2.37	0.718
**Communication skill** (adequate vs. inadequate #)	7.92	2.15, 29.24	0.002 *
Decision skill/ (adequate vs. inadequate #)	6.00	1.86, 19.36	0.003 *
**Self-management skill** (adequate vs. inadequate #)	2.95	1.08, 8.11	0.036 *

# Reference group, * *p*-value < 0.05, OR adjusted by age and sex.

## Data Availability

Not applicable.

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
