# Peer review of "Correctional Officers’ Health Literacy and Practices for Pulmonary Tuberculosis Prevention in Prison"

_ijerph, 2022, doi:10.3390/ijerph191811297_

Round 1
Reviewer 1 Report
The present study is pertinent, but methods and study findings must be clarified and intelligibly presented. Methods and study findings are not clearly presented (please see my commentaries below). Crucial concepts are not explained. The original versions of the administered tools are not available in a supplementary file.
Abstract
What is a correction officer? Please briefly define all concepts when first introduced in the paper.
“under the limitations of health professionals in prisons (e.g., a, b, …)”; please present some examples of limitations.
“HL and practices self-administered questionnaires were used for data collection” HL self-administered questionnaire (ok) vs practices (?) self-administered questionnaire? Please give more details.
“Findings 24 support that COs have adequate HL, and is associated with good practices in TB prevention” or “Findings 24 support that COs have adequate HL, which is associated with good practices in TB prevention”? The paper must be proofread by an English native speaker.
Keywords: please use the maximum number of keywords according to instructions for authors. Please use some MeSH terms. Please select different keywords/words from those used in abstract.
Introduction
- “Tuberculosis (TB) is a contagious disease that constitutes a significant public health 31 problem and causes morbidity and mortality worldwide.” Please give more details about TB? Etiology? Prognosis? Prevention measures? Specific prevention measures in prisons? Treatment (pharmacologic vs. non-pharmacologic?
- Please describe the 5 most TB prevalent countries and compare developed vs. developing countries.
- How many annual deaths due to TB? Economic costs?
- Please present the full meaning of abbreviations when first presented in the text.
- Line 57: please explain what is a correctional officer? Please define all concepts when first introduced in the text.
- Lines 60-64: references are missing. All sentences require references in introduction, but not in discussion. Please check the presentation of references in introduction.
- Line 66: “End TB and SDGs” Why is “End” capitalized? Please proofread the paper. Please present the full meaning of all abbreviations when first presented in the text.
- Line 65: “Providing advocacy, and partnerships is one of the strategies that WHO suggested for End TB and SDGs for ending the TB epidemic by 2030 [17].” Please give more details about this sentence. Advocacy? And Partnerships? Additionally, please describe some practical examples.
- Lines 65-70: references are missing.
- Lines 65 to 80: please create at least one additional paragraph.
- “However, few studies have investigated HL of the CO on TB prevention and control in prisons.” References are missing. Have authors carried out a systematic review? Please consider rephrase the present sentence.
2. Materials and Methods
In general, materials and methods are too vague and more details must be provided.
2.1. Study design and participants
- Line 91: “in five prisons of Nakhon Si Thammarat, Thailand.” Name of prisons?
- Line 93-94: “sample size that was calculated by using the finite population proportion formula with p=0.5 was 193”. References are missing. Preferably, present the formula and the used values.
- Lines 95-96: “Stratified random sampling proportional to the size of each prison was used to select the participants.” References are missing. Please give more details.
2.2. Data collection and measurements
- A questionnaire or two questionnaires? Please see abstract.
- Please present the original version(s) in a supplementary file?
- How was the questionnaire(s) developed?
- How was the questionnaire(s) pre-tested?
- How was the questionnaire(s) validated?
2.3. HL assessment and 2.4. Practices for TB prevention and control in prison
- The original version of all self-administered tools/questionnaires should be presented in a supplementary file.
2.4. Practices for TB prevention and control in prison
- References are missing.
2.5. Data processing and analysis
The data collected were entered into a statistical software package for processing and analysis.
-Name, version, and reference of the software package?
- Please present the scoring methodology in the section where the respective tool/questionnaire is described.
- Please describe here the applied statistical methods. References are missing. Please cite studies applying similar statistical methodologies.
- Why was a multivariate logistic regression model not calculated?
- I recommend the calculation of a multivariate logistic regression model. Please see https://www.sciencedirect.com/topics/medicine-and-dentistry/multivariate-logistic-regression-analysis
3. Results
- Please check the presentation of statistical finding in APA guidelines.
- 3.1. Description of the participants
- Please place this information in methods: “one central prison and four local prisons of the province or present here the number of participants per each prison.
- Table 1: preferably do not use abbreviations in Tables (or obligatory present their full meaning below the Table).
- Please check Tables and Figures formats in instructions for authors and/or in published papers.
3.2. HL for TB prevention and control
- “In total, 64 (30.8%) had an adequate level of knowledge.” Please update the subheading 3.2. For instance “HL for TB prevention and control and participants’ knowledge on TB” (or other).
- Please note that knowledge and HL are not necessary the same.
- Please also update “2.3. HL assessment”. For instance, “2.3. HL assessment and participants’ knowledge on TB” (or other). Please unequivocally identify the tools/question used to evaluate HL and knowledge, respectively in methods.
3.2. HL for TB prevention and control 174
- “Nearly all participants had an adequate level of total HL (n=131; 63.0%).” Is 63% nearly all? Please rephrase the sentence.
Discussion
- Please cite more related or similar studies.
- Please cite some systematic reviews on similar or related topics.
- Please be careful with this type of sentences “This study is the first to enroll CO in Thailand for a HL study.”
- “The majority of the participants (63%) had adequate HL for TB prevention and control. Approximately 70% of the participants had inadequate cognitive skills.” Are these data congruent? Please discuss. How someone who is not cognitive competent present an adequate health literacy?
- The study participants were the COs? Please describe the roles of COs in prisons (add this information in study introduction). What are the tasks of COs in prisons? “Approximately 70% of the participants had inadequate cognitive skills.” What are inadequate cognitive skills? How is this possible? Why almost all correctional officer (around 70%) had inadequate cognitive skills? Why are they not dismissed? Please unequivocally clarify these issues in methods: “The HL assessment included six aspects, including 1) cognitive skill; 2) access skill; 3) communication skill; 4) decision skill; 5) self-management skill; and 6) media literacy, that were captured using 41 questions.” Please unequivocally present the questions per each one of the five dimensions in methods, and preferably repeat them in discussion (or cite the section of the questionnaire in the supplementary file).
- Please not that by definition “Cognitive skills are the core skills your brain uses to think, read, learn, remember, reason, and pay attention. Working together, they take incoming”.
- Please unequivocally discuss all evaluated dimensions i.e., 1) cognitive skill; 2) access skill; 3) communication skill; 4) decision skill; 5) self-management skill; and 6) media literacy.
- Please not present study findings (e.g., %s or proportions) in discussion.
- What was the contribute of study authors for improving the health/health literacy/knowledge of prisoners in the evaluated prisons?
- Please create a section on practical implication (what are the study strengths and why the present study is relevant to an international audience) and future research at the end of discussion.
Conclusion
- Conclusion must reply to study objectives.
- Please do not place practical implications in study conclusions.
References
- Please check the format of references in instructions for authors.
Author Response
Reviewer 1 response
Dear Reviewer
Re: Correction Officers’ Health Literacy and Practices for Pulmo-nary Tuberculosis Prevention in Prison
Thank you very much for giving us an opportunity to revise this important piece of work. We have done the revision, taking into account all comments. Our responses are set out below.
The present study is pertinent, but methods and study findings must be clarified and intelligibly presented. Methods and study findings are not clearly presented (please see my commentaries below). Crucial concepts are not explained. The original versions of the administered tools are not available in a supplementary file.
Point1
Abstract
What is a correction officer? Please briefly define all concepts when first introduced in the paper.
“under the limitations of health professionals in prisons (e.g., a, b, …)”; please present some examples of limitations.
“HL and practices self-administered questionnaires were used for data collection” HL self-administered questionnaire (ok) vs practices (?) self-administered questionnaire? Please give more details.
“Findings 24 support that COs have adequate HL, and is associated with good practices in TB prevention” or “Findings 24 support that COs have adequate HL, which is associated with good practices in TB prevention”? The paper must be proofread by an English native speaker.
Response to point 1: We have explained more clear details and proofread by an English native speaker.
Point 2
Keywords: please use the maximum number of keywords according to instructions for authors. Please use some MeSH terms. Please select different keywords/words from those used in abstract.
Response to point 2: We have done.
Point 3
Introduction
- “Tuberculosis (TB) is a contagious disease that constitutes a significant public health 31 problem and causes morbidity and mortality worldwide.” Please give more details about TB? Etiology? Prognosis? Prevention measures? Specific prevention measures in prisons? Treatment (pharmacologic vs. non-pharmacologic?
- Please describe the 5 most TB prevalent countries and compare developed vs. developing countries.
- How many annual deaths due to TB? Economic costs?
- Please present the full meaning of abbreviations when first presented in the text.
- Line 57: please explain what is a correctional officer? Please define all concepts when first introduced in the text.
- Lines 60-64: references are missing. All sentences require references in introduction, but not in discussion. Please check the presentation of references in introduction.
- Line 66: “End TB and SDGs” Why is “End” capitalized? Please proofread the paper. Please present the full meaning of all abbreviations when first presented in the text.
- Line 65: “Providing advocacy, and partnerships is one of the strategies that WHO suggested for End TB and SDGs for ending the TB epidemic by 2030 [17].” Please give more details about this sentence. Advocacy? And Partnerships? Additionally, please describe some practical examples.
- Lines 65-70: references are missing.
- Lines 65 to 80: please create at least one additional paragraph.
- “However, few studies have investigated HL of the CO on TB prevention and control in prisons.” References are missing. Have authors carried out a systematic review? Please consider rephrase the present sentence.
Response to point 3: We have revised, explained more details of TB epidemic and its impact, correctional officers’ definition and duties, and added more references.
Point 4
- Materials and Methods
In general, materials and methods are too vague and more details must be provided.
Response to point 4: We have provided more details.
Point 5
2.1. Study design and participants
- Line 91: “in five prisons of Nakhon Si Thammarat, Thailand.” Name of prisons?
- Line 93-94: “sample size that was calculated by using the finite population proportion formula with p=0.5 was 193”. References are missing. Preferably, present the formula and the used values.
- Lines 95-96: “Stratified random sampling proportional to the size of each prison was used to select the participants.” References are missing. Please give more details.
Response to point 5: We have done according to the suggestions.
Nakhon Si Thammarat is the name of a province. We have provided more details.
Add sample size determination reference. (Cochran, W. G. (1977). Sampling techniques. 3rd Ed. New York: John Wiley & Sons.)
Add sampling technique reference.(Suresh K, Thomas SV, Suresh G. Design, data analysis and sampling techniques for clinical research. Ann Indian Acad Neurol. 2011 Oct;14(4):287-90. doi: 10.4103/0972-2327.91951. PMID: 22346019; PMCID: PMC3271469.)
Point 6
2.2. Data collection and measurements
- A questionnaire or two questionnaires? Please see abstract.
- Please present the original version(s) in a supplementary file?
- How was the questionnaire(s) developed?
- How was the questionnaire(s) pre-tested?
- How was the questionnaire(s) validated?
Response to point 6:
We developed a questionnaire (four parts) by ourselves using the health literacy six dimensions by applying the concept of Nutbeam (2002) and reference no. 29.
We have done the content validity test and reliability test (pre-test) as shown in topic 2.2.
Point 7
2.3. HL assessment and 2.4. Practices for TB prevention and control in prison
- The original version of all self-administered tools/questionnaires should be presented in a supplementary file.
Response to point 7: We developed a questionnaire (four parts) by ourselves using the health literacy six dimensions by applying the concept of Nutbeam (2002, 2008) and the Department of Health Service Support 2020 (reference no. 30). Reference no.30 could access via the link by a reference list (in Thai).
Point 8
2.4. Practices for TB prevention and control in prison
- References are missing.
Response to point 8: We have done by ref. no. 15.
Point 9
2.5. Data processing and analysis
The data collected were entered into a statistical software package for processing and analysis.
-Name, version, and reference of the software package?
- Please present the scoring methodology in the section where the respective tool/questionnaire is described.
We have done in topic 2.5. (Epi data version 3.1 for cross-checking and exported to SPSS version 23.0 for analysis).
- Please describe here the applied statistical methods. References are missing. Please cite studies applying similar statistical methodologies.
We have added the reference. (Feleke BT, Wale MZ, Yirsaw MT (2021) Knowledge, attitude and preventive practice towards COVID-19 and associated factors among outpatient service visitors at Debre Markos compressive specialized hospital, north-west Ethiopia, 2020. PLoS ONE 16(7): e0251708. https://doi.org/10.1371/journal.pone.0251708)
- Why was a multivariate logistic regression model not calculated?
- I recommend the calculation of a multivariate logistic regression model. Please see https://www.sciencedirect.com/topics/medicine-and-dentistry/multivariate-logistic-regression-analysis
This study also used the multivariate logistic regression with Binary logistic regression (LR) analysis, which is a regression model where the target variable is binary (https://www.sciencedirect.com/topics/computer-science/binary-logistic-regression#:~:text=Binary%20logistic%20regression%20(LR)%20is,or%20not%20readmitted%20(0).)
Response point 9: We have done.
Point 10
- Results
- Please check the presentation of statistical finding in APA guidelines.
- 3.1. Description of the participants
- Please place this information in methods: “one central prison and four local prisons of the province or present here the number of participants per each prison.
- Table 1: preferably do not use abbreviations in Tables (or obligatory present their full meaning below the Table).
- Please check Tables and Figures formats in instructions for authors and/or in published papers.
Response to point 10: We have checked and placed abbreviations with complete words in table 1.
Point 11
3.2. HL for TB prevention and control
- “In total, 64 (30.8%) had an adequate level of knowledge.” Please update the subheading 3.2. For instance “HL for TB prevention and control and participants’ knowledge on TB” (or other).
- Please note that knowledge and HL are not necessary the same.
- Please also update “2.3. HL assessment”. For instance, “2.3. HL assessment and participants’ knowledge on TB” (or other). Please unequivocally identify the tools/question used to evaluate HL and knowledge, respectively in methods.
Response point 11: We have done (Knowledge was assessed for the cognitive skill of health literacy. Thus, it has been included in the health literacy assessment. We have explained more details on the health literacy assessment and questionnaire in topic 2.3.).
Point 12
3.2. HL for TB prevention and control 174
- “Nearly all participants had an adequate level of total HL (n=131; 63.0%).” Is 63% nearly all? Please rephrase the sentence.
Response to point 12: We have done.
Point 13
Discussion
- Please cite more related or similar studies.
- Please cite some systematic reviews on similar or related topics.
- Please be careful with this type of sentences “This study is the first to enroll CO in Thailand for a HL study.”
We have rewritten the sentence.
- “The majority of the participants (63%) had adequate HL for TB prevention and control. Approximately 70% of the participants had inadequate cognitive skills.” Are these data congruent? Please discuss. How someone who is not cognitive competent present an adequate health literacy?
- The study participants were the COs? Please describe the roles of COs in prisons (add this information in study introduction). What are the tasks of COs in prisons? “Approximately 70% of the participants had inadequate cognitive skills.” What are inadequate cognitive skills? How is this possible? Why almost all correctional officer (around 70%) had inadequate cognitive skills? Why are they not dismissed? Please unequivocally clarify these issues in methods: “The HL assessment included six aspects, including 1) cognitive skill; 2) access skill; 3) communication skill; 4) decision skill; 5) self-management skill; and 6) media literacy, that were captured using 41 questions.” Please unequivocally present the questions per each one of the five dimensions in methods, and preferably repeat them in discussion (or cite the section of the questionnaire in the supplementary file).
- Please not that by definition “Cognitive skills are the core skills your brain uses to think, read, learn, remember, reason, and pay attention. Working together, they take incoming”.
- Please unequivocally discuss all evaluated dimensions i.e., 1) cognitive skill; 2) access skill; 3) communication skill; 4) decision skill; 5) self-management skill; and 6) media literacy.
- Please not present study findings (e.g., %s or proportions) in discussion.
- What was the contribute of study authors for improving the health/health literacy/knowledge of prisoners in the evaluated prisons?
- Please create a section on practical implication (what are the study strengths and why the present study is relevant to an international audience) and future research at the end of discussion.
Response point 13: We have done in discussion part.
Point 14
Conclusion
- Conclusion must reply to study objectives.
- Please do not place practical implications in study conclusions.
Response point 14: We have done.
Point 15
References
- Please check the format of references in instructions for authors.
Response point 15: The journal will edit after accepted.

Reviewer 2 Report
Dear Authors!
Since this is the first paper on the health literacy pf correction officers and its implications for the health of prisoners in Thailand, the study has several merits. Data collection and measurement procedures are clearly presented and understandable to the reader. However, there are some considerations I need to address in order for the paper to be worth publishing.
1. The biggest deficiency is the use of the measure History of training on epidemics for descriptive purposes only. This crucial feature of a correction officer should appear as an explanatory variable in its own right, provided that the logic of the paper is build around the advocacy potential of these officers for the proper health conduct of prisoners. Education is being used in the logistic regression model, but why is History on training ... involved at all, if only used for a description in one single table? If such a training/search for information of officers is irrelevant, than they cannot be suitable advocates of change. Please refer to this issue at least in the Discussion, but even better in the Result section by including the measure in the logistic regression model.
2. The part in lines 245-250 rather belongs to the introductory part, as the Discussion should not contain novel elements.
3. The Discussion barely has any other research results to refer to. Even though this was the first assessment of the correction officers' health literacy in Thailand, there must be other research results to provide some context for the discussion. Please search for some international reference that underscore the logic of argumentation.
4. In section 2.3 HL assessment is quite clear, but the formulation "truth about yourself" (line 115) is not clear. What are the remaining five skil ls? Of how many skills did these 5 skills remain? The reader does not understand clearly (line 117). Later on, in Table 1, the same issue appears as HISTORY of training on epidemic prevention and control, I suggest using this notion in part 2.3 too. A little bit more clarification on all measurement tools is welcome, as a total of 4 notions were measured: HL, cognitive knowledge on TB, history of training and practices for TB prevention and control. Further, it does not become clear whether the questions were taken from existing questionnaires or formulated by the authors.
5. The cut-off points are clearly described, however, the foundations laid by previous studies cannot be found, the link provided to reference no. 23 is not existing and there is no way to search for reference no. 24, as the book is not available online. Please provide further evidence for the cut-off points used for statistical analysis, otherwise this argument remains very weak. Also, the measurement of the six aspects of HL remains somewhat unclear.
6. With respect to linguistic issues and style, there are some minor issues to be dealt with. In some sentences the abbreviations CO HL and TB are too much, too dense, like lines 81-88. For example, CO's HL is not adequate for a research paper. I suggest writing the whole term and only use one abbreviation in a sentence, the most.
Line 39: ASEAN countries (probably Asian would be correct).
Line 110: .... age in the village of health behaviour change 2020 - this line is very fuzzy, please rephrase.
I suggest replacing the notion "demographics" to sociodemographic variables.
Lines 190-191: .... was only 34.1% for the extremely practices - please rephrase.
Line 197: good practices in f each HL dimension - possible mistyping
Lines 280-282: The sentence has no predicate.
Good luck with the work on the paper.
Best wishes,
reviewer
Author Response
Reviewer 2 response
Dear Reviewer
Re: Correction Officers’ Health Literacy and Practices for Pulmo-nary Tuberculosis Prevention in Prison
Thank you very much for giving us an opportunity to revise this important piece of work. We have done the revision, taking into account all comments. Our responses are set out below.
Point 1: The biggest deficiency is the use of the measure History of training on epidemics for descriptive purposes only. This crucial feature of a correction officer should appear as an explanatory variable in its own right, provided that the logic of the paper is build around the advocacy potential of these officers for the proper health conduct of prisoners. Education is being used in the logistic regression model, but why is History on training ... involved at all, if only used for a description in one single table? If such a training/search for information of officers is irrelevant, than they cannot be suitable advocates of change. Please refer to this issue at least in the Discussion, but even better in the Result section by including the measure in the logistic regression model.
Response point 1: The training history variable is one of the participant characteristics. We have analyzed the correlation (bivariate analysis) of training history and practice with Pearson’s Chi-square test. The result shows in Table 3 with a p-value is 0.217. Thus, this variable could not select for the logistic regression analysis according to the variable selection criteria stated in the statistical analysis.
Point 2: The part in lines 245-250 rather belongs to the introductory part, as the Discussion should not contain novel elements.
Response point 2: We have removed (line 247-249).
Point 3: The Discussion barely has any other research results to refer to. Even though this was the first assessment of the correction officers' health literacy in Thailand, there must be other research results to provide some context for the discussion. Please search for some international reference that underscore the logic of argumentation.
Response point 3: We have done
Point 4: In section 2.3 HL assessment is quite clear, but the formulation "truth about yourself" (line 115) is not clear. What are the remaining five skills? Of how many skills did these 5 skills remain? The reader does not understand clearly (line 117). Later on, in Table 1, the same issue appears as HISTORY of training on epidemic prevention and control, I suggest using this notion in part 2.3 too. A little bit more clarification on all measurement tools is welcome, as a total of 4 notions were measured: HL, cognitive knowledge on TB, history of training and practices for TB prevention and control. Further, it does not become clear whether the questions were taken from existing questionnaires or formulated by the authors.
Response point 4: We developed the questionnaire according to the six HL domain, including 1) cognitive skill; 2) access skill; 3) communication skill; 4) decision skill; 5) self-management skill; and 6) media literacy, that were captured using 41 questions. The questions on cognitive skills pertaining to knowledge of TB infection were structured, pre-coded, and mainly evoked dichotomous responses (“yes” or “no”). For the remaining five skills, participants self-reported HL provided by the five-point response scale. The training history variable is one of the characteristics variable that we included in the participants characteristics. (We have explained in the section 2.3)
Point 5: The cut-off points are clearly described, however, the foundations laid by previous studies cannot be found, the link provided to reference no. 23 is not existing and there is no way to search for reference no. 24, as the book is not available online. Please provide further evidence for the cut-off points used for statistical analysis, otherwise this argument remains very weak. Also, the measurement of the six aspects of HL remains somewhat unclear.
Response point 5: Reference no. 23 was in Thai and has been found via the link. We have changed reference list no. 24. [Akalu Y, Ayelign B, Molla MD: Knowledge, attitude and practice towards COVID-19 among chronic disease patients at Addis Zemen Hospital, Northwest Ethiopia. Infection and drug resistance 2020, 13:1949.]
Point 6: With respect to linguistic issues and style, there are some minor issues to be dealt with. In some sentences the abbreviations CO HL and TB are too much, too dense, like lines 81-88. For example, CO's HL is not adequate for a research paper. I suggest writing the whole term and only use one abbreviation in a sentence, the most.
Response point 6: We have done.
Point 7: Line 39: ASEAN countries (probably Asian would be correct).
Response point 7: We have corrected (line 39).
Point 8: Line 110: .... age in the village of health behaviour change 2020 - this line is very fuzzy, please rephrase.
Response point 8: We have done (line 109-111).
Point 9: I suggest replacing the notion "demographics" to sociodemographic variables.
Response point 9: We have done (line 207).
Point 10: Lines 190-191: .... was only 34.1% for the extremely practices - please rephrase.
Response point 10: We have done ( line 191-192).
Point 11: Line 197: good practices in f each HL dimension - possible mistyping
Response point 11: We have corrected (line 197).
Point 12: Lines 280-282: The sentence has no predicate.
Response point 12: We have corrected (line 280-283).

Round 2
Reviewer 1 Report
Dear authors, the quality of the paper has been improved. Congratulations.
I highly recommend the presentation of the original versions of all administered tools in a supplementary file/annex. Materials and methods must be reproductible per se.
Minor revisions:
Please see point 2.3 "HL assessment about TB prevention and control in prisons was undertaken with a 134 tool that was constructed by the authors using the concept of six aspects of HL skills (Annex A or supplementary file A)",
Please see point 2.4 "We asked the participants about what they were doing or would do to protect them- 147 selves from (16 items) and control TB transmission (12 items) Annex B or supplementary file B)",
- "six aspects of HL skills. [25, 135 29-30]" or "six aspects of HL skills [25, 135 29-30]." ?
Please proofread the paper once more.
Author Response
Reviewer#1
I highly recommend the presentation of the original versions of all administered tools in a supplementary file/annex. Materials and methods must be reproductible per se.
Minor revisions:
Dear Reviewer
Re: Correction Officers’ Health Literacy and Practices for Pulmonary Tuberculosis Prevention in Prison
Thank you very much again for giving us an opportunity to revise this important piece of work. We have done the revision, taking into account all comments. Our responses are set out below.
Point1
Please see point 2.3 "HL assessment about TB prevention and control in prisons was undertaken with a 134 tool that was constructed by the authors using the concept of six aspects of HL skills (Annex A or supplementary file A)",
Response to point1
The questionnaire was created in Thai language. Therefore, there may be restrictions on use. We are happy to give it to interested readers by contacting the author directly.
Point2
Please see point 2.4 "We asked the participants about what they were doing or would do to protect them- 147 selves from (16 items) and control TB transmission (12 items) Annex B or supplementary file B)",
Response to point2
The questionnaire was created in Thai language. Therefore, there may be restrictions on use. We are happy to give it to interested readers by contacting the author directly.
Point3
- "six aspects of HL skills. [25, 135 29-30]" or "six aspects of HL skills [25, 135 29-30]." ?
Response to point3
We have corrected it (line135).
Point4
Please proofread the paper once more.
Response to point4
We have done.

Reviewer 2 Report
Dear Authors!
Thanks for providing the corrections and additions in such a short time.
Congratulations for the improved quality of this paper.
Best,
reviewer
Author Response
Reviewer#2
Dear Authors!
Thanks for providing the corrections and additions in such a short time.
Congratulations for the improved quality of this paper.
Best,
Reviewer
Response to reviewer
Dear Reviewer
Re: Correction Officers’ Health Literacy and Practices for Pulmonary Tuberculosis Prevention in Prison
Thank you very much again for giving us an opportunity to revise this important piece of work.
